# Prognostic Value of Coronary Calcium Score in Asymptomatic Individuals: A Systematic Review

**DOI:** 10.3390/jcm11195842

**Published:** 2022-10-01

**Authors:** Liberatore Tramontano, Bruna Punzo, Alberto Clemente, Sara Seitun, Luca Saba, Eduardo Bossone, Erica Maffei, Carlo Cavaliere, Filippo Cademartiri

**Affiliations:** 1IRCSS SYNLAB SDN, 80143 Naples, Italy; 2Department of Radiology, Fondazione Monasterio/CNR, 56124 Pisa, Italy; 3Department of Radiology, IRCCS Ospedale San Martino, 16132 Genoa, Italy; 4Department of Radiology, University of Cagliari, 09100 Cagliari, Italy; 5Department of Cardiology, Azienda Ospedaliera Antonio Cardarelli, 80131 Naples, Italy

**Keywords:** calcium score, primary prevention, agatston, risk assessment, coronary artery disease

## Abstract

Despite updated guidelines and technological developments that allow for an accurate diagnosis, many asymptomatic individuals have a high risk of developing CAD or cardiac events. The CAC score can estimate a correct risk level for these subjects, which is clinically significant for adequate management of risk factors and obtaining personalized preventive therapy. This systematic review aims to assess the prognostic value of CAC score in asymptomatic individuals. According to the Preferred Reporting Items for Systematic Reviews and Meta-Analyses (PRISMA) statement, a systematic literature search was performed to identify original articles since 2010 that evaluated the prognostic value of CAC score in asymptomatic individuals. The quality of the included studies was assessed by the QUIPS tool. A total of 45 articles were selected. Many of these (25 studies) evaluated the prognostic value of CAC score in asymptomatic subjects. In comparison, others (20 studies) evaluated the association of CAC score with other clinical parameters and imaging modalities or the comparison with computed tomography coronary angiography (CTCA). Our findings showed that the CAC score provides valuable prognostic information for predicting CAD risk in asymptomatic individuals.

## 1. Introduction

Cardiac computed tomography (CCT) can assess the presence of calcium on the coronary tree and estimate the risk of coronary artery disease (CAD). Cardiovascular risk of events can be estimated as coronary artery calcium (CAC) score. CAC score is a non-invasive, accessible, fast, reliable, and reproducible method [1,2] that allows for the gathering of direct information about coronary atherosclerosis without injection of iodinated contrast medium. It provides information about actual cardiovascular risk and its implication for optimal medical therapy in primary prevention and future development of CAD complications (major adverse coronary events, MACE) [3]. Few studies have evaluated the predictive power of CAC score in symptomatic patients [4]. However, most of the literature has been focused on asymptomatic individuals; in this scenario, a CAC score would play a significant role in driving primary prevention strategies to prevent future cardiac events [5]. Several other uses of the CAC score have been assessed. Argawal et al. [6] evaluated the predictive value of CAC score in 1051 patients with type 2 diabetes for approximately 7.4 years, demonstrating a higher mortality risk for patients with a CAC ≥ 100 (20% compared to 6.7% for CAC = 0). Moreover, Argawal et al. [7] demonstrated a higher predictive value of the CAC score than the Framingham risk score (FRS), determining a reclassification of cardiovascular risk in about 25% of patients and allowing more delicate clinical management of patients undergoing CCT angiography (CTCA) [8,9]. Similar changes in cardiovascular risk assessment have been reported by other studies, ranging from 21% to 52% [5,6,7,8,9,10,11].

In other cases, the authors evaluated CAC score thresholds to decide the optimal follow-up timing for asymptomatic patients [12] and assessed its value in different ethnic groups [13,14,15,16,17]. For instance, in their study, Orimoloye et al. [17], which included 38,277 whites, 1621 Asians, 977 blacks, and 1349 Hispanics, followed for 11.7 years, showed that the CAC score represents an excellent predictive tool for all-cause and cardiovascular disease (CVD) mortality. CAC score was higher in black and Hispanic patients, placing them at greater risk than in Asians and whites. These studies suggested improving the basic guidelines, pointing out the role of inter-racial differences in terms of biological or social factors, leading to different risk stratification and prognosis. Finally, the CAC score has also been associated with other biomarkers to build a composite multivariable index for risk stratification.

Many have seen that CAC score increased in association with the number of plaques [18], the albumin value [19], or N-terminal pro-brain natriuretic peptide [20] in asymptomatic type 2 diabetic patients. Kemmer et al. [21] used it as a CAD screening tool in liver transplant patients, while Park et al. [22] found that smokers with low LDL are strongly associated with obstructive CAD.

Several drawbacks affect the CAC score regarding mortality risk estimation. Cademartiri et al. [23] showed that the CAC score, when used alone in high-risk asymptomatic individuals, has a high sensitivity (91.9%). However, low specificity (75%) and a high percentage of false positives (about 21%) resulted from inadequate detection of obstructive or non-obstructive CAD compared to CTCA. Similarly, Han et al. [24] confirmed that CTCA is better than CAC score only as a risk predictor, and Dedic et al. [25] showed how CTCA is helpful in the reclassification of those patients that have CAC = 0, allowing, at the same time, better visualization of the stenosis and the severity in patients affected by CAD. Another secondary disadvantage could be the radiation dose provided to perform CT in asymptomatic patients. Some studies have evaluated the impact of different acquisition protocols (e.g., varying kiloVolt [26], kernel filters [27]), demonstrating a dose reduction of about 80%. However, in these cases, the feasibility of a different, non-standardized CAC protocol has to be reassessed, even longitudinally.

This systematic review aims to collect, summarize, and discuss recent studies evaluating the role of the CAC score as a prognostic tool in asymptomatic patients.

## 2. Materials and Methods

### 2.1. Search Strategy and Selection Criteria

A systematic search for all the published studies examining the prognostic value of CAC score in asymptomatic individuals was conducted. The most relevant electronic databases (PubMed, Web of Science, and Google Scholar) were comprehensively explored and used to build the search. Only studies published from 2010 to October 2021 were selected. The search strategy included the key terms listed in Appendix A.

The literature search was restricted to English language publications and studies of human subjects. After having screened identified titles and abstracts, two reviewers assessed the full text of the articles that evaluated the prognostic value of CAC score in asymptomatic individuals and were not review articles. For articles with full-text available criteria, further selection criteria had to be fulfilled: asymptomatic patients; patients with CAC score; and CAC score as a prognostic tool. Studies were excluded if the patient population included people with symptoms such as chest pain or were used for other types of scores other than the CAC score.

### 2.2. Planning and Conducting the Review

After the selection procedure, the selected articles were analyzed by two reviewers, and data helpful in conducting the systematic review were collected in a predesigned sheet. The extracted data included the following: study characteristics (first author name, publication year and study design prospective/retrospective and number of patients); patient characteristics (age, symptomatic/asymptomatic, risk factors, time of follow-up); type of imaging (EBCT or CT); acquisition details (CT protocol where described); statistical analysis; and main findings. The articles were classified and analyzed according to the CAC score as a prognostic tool. If the CAC score was associated with other techniques such as CCTA or SPECT in the same study, they were evaluated as belonging to a separate study. This systematic review was conducted following the Preferred Reporting Items for Systematic Reviews and Meta-Analyses (PRISMA) statement (see Appendix A for the PRISMA checklist) [28].

### 2.3. Quality Assessment

The quality of the individual studies was assessed using the Quality in Prognosis Studies (QUIPS) tool [29,30]. According to QUIPS, six domains are critical for assessing biases in prognostic studies: selection of study participants, study attrition, prognostic factor measurement, outcome measurement, study confounding, and statistical analysis and reporting. For each of these six domains, the responses “yes”, “partial”, “no”, or “unsure” for three up to seven items within each domain are combined to assess the risk of bias. An overall rating for each domain is assigned as “high”, “moderate”, or “low” risk of bias. Two reviewers independently completed the QUIPS assessment for each study, and the discussion resolved discrepancies.

## 3. Results

### 3.1. Study Selection

A total of 222 articles were retrieved by the scientific electronic databases search. Eleven additional articles were found through article references, bringing the total number of records suitable for further evaluation to 233. After removing the duplicates, there were 179 articles left for investigation. By scanning the title and abstract of these records, 28 records were excluded because they did not match the inclusion criteria (23 review articles, 2 case reports, 1 meta-analysis, 1 clinical trial, and 1 article in the French language). A total of 151 articles were evaluated on their full text. Of these articles, 106 records were excluded on the basis of the inclusion criteria (96 were not on CAC score as a prognostic tool, 5 were only on CTCA, and 5 included symptomatic patients). Finally, 45 records were included for qualitative synthesis. The PRISMA flow diagram of the included studies according to the inclusion and exclusion criteria is reported in Figure 1.

### 3.2. Characteristics of the Included Studies

The selected studies on CAC score are 45. All the selected studies were targeted at adults, and the median number of individuals was 13,902. The mean age of patients was 52.66 years. A part of the study design was retrospective (15/45). Most of the selected papers (28 studies) involved patients without symptoms.

Five studies included asymptomatic patients with diabetes; four did not specify if patients were asymptomatic or not; and one also included patients with symptoms and others with ischemic patients. Twenty studies evaluated the CAC score as a prognostic tool. For most patients, it was found that developing a cardiac event or death was related to high values of the CAC score, whereby patients with low or no CAC score had an improved long-term prognosis.Some of the selected papers examined the association of the CAC score with other methods, parameters, or biomarkers. There were studies (3/45) that evaluated the combination of information obtained from the CAC score and the SPECT or associated with the CTCA to make a better prognosis in asymptomatic patients; others (6/45) evaluated the association of NT-proBNP with coronary flow velocity reserve. Concerning what could be the prognostic limits of the CAC score, there were some studies (11/45) that highlighted the advantages of CTCA over the CAC score in asymptomatic patients.

### 3.3. Prognostic Value of Coronary Calcium Score

#### 3.3.1. Coronary Calcium Score

A total of 20 studies investigated the predictive power of the CAC score in asymptomatic patients. Seven evaluated the CAC score to define the risk of developing cardiovascular events or diseases; two studies were on the predictive power of CAC = 0 in the long term. Four evaluated the CAC score associated with risk factors such as smoking, hypertension, and familiarity for CAD. The other two evaluated the warranty period within which the predictive power of the CAC score could fall; two looked for score differences between different races/ethnicities or simply between males and females; and three evaluated some aspects of the CAC score, such as the Coronary Artery Calcium Data and Reporting System (CAC-DRS).

Dudum et al. [31] verified the recommendations of the society of cardiovascular computed tomography (SCCT) in using CAC score in patients with familiarity with coronary heart disease (CHD) and with atherosclerotic cardiovascular disease (ASCVD) risk < 5%. In 14,169 individuals followed some for 11 and other 6 years, the risk rate was three times higher in patients with CAC > 100 (particularly 4.7 times for CVD, 11.4 times for CHD) than in those with low or 0 CAC. The results demonstrated the reliable prognostic power beneficial for low-risk patients who need more aggressive therapy, as Cho et al. [32] deemed appropriate. The authors demonstrated that patients with CAC > 100 and with obstructive coronary stenosis had 11.6% more mortality risk than those with low CAC and without stenosis, whose risk was 2.9% during five years of follow-up.

The age for which elderly patients are most at risk may also be an essential factor, but the study results by Carr et al. [33] found something else. They evaluated the predictive power of the CAC score in predicting the risk of CHD and CVD in patients aged 32–46 years during a 12.5-year follow-up. The risk for those with CAC > 100 was found to be 3.7-fold higher than for those with low or 0 scores. Similar findings were highlighted by Han et al. [34] that demonstrated how CAC score was independently associated with all-cause mortality, and the risk for patients with CAC > 400 was 2.3 times higher than those with CAC = 0. The CAC score allowed a better definition of the risk than other models, e.g., FRS [35]. The risk of death was more significant for patients with CAC ≥ 400 and without risk factors than for those with one or more risk factors but without CAC, suggesting that a predictive model based only on traditional risk factors cannot be used. The risk estimate provided by the CAC score is reliable even after 15 years, as found by Shaw et al. [36].

Other authors focused on identifying threshold values beyond which there could be a certain risk of death. In the study of Patel et al. [37], 10-year survival was 99% for subjects with CAC = 0, 74–78% for those with CAC > 1000, and 51% for patients with CAC > 2000. The risk for those with a CAC between 1000 and 2000 was similar. However, it was clear that patients with a score higher than 2000 had a greater certainty of being able to have a cardiac event, also due to the extension of calcified plaques and clinical complications that could develop. Since there was no well-defined threshold value, the attention of the studies has turned to how long the warranty period could be for those subjects with CAC = 0 whose prognosis, as demonstrated by other studies, is favorable compared to those with any CAC score. Valenti et al. [38] evaluated the warranty period of asymptomatic patients with CAC = 0 by comparing it to other predictive models such as FRS and adult treatment panel III (NCEP ATP III). The warranty period was defined as when the subject’s risk did not change for which it remained classified as low risk; when the risk changed, the warranty period ended. The results showed that the warranty period for those subjects with CAC = 0 was 15 years instead of for people with age ≥ 60 years, but with CAC = 0, it was slightly reduced; these subjects had more prolonged survival than those considered low/intermediate risk in the presence of any CAC score. Similarly, Lee et al. [39] found that patients with CAC = 0 had a more extended warranty period than subjects with CAC > 0 and, at the same time, highlighted the reliability of the CAC score as an excellent prognostic tool. Blaha et al. [40] assessed how cardiac events, or in some cases death, occurred in subjects with a low CAC score or equal to 0. There was no doubt that the CAC score was a reliable tool; in fact, the risk for those with CAC > 10 was four times higher than those with CAC = 0, but the most surprising thing was that in patients with low or 0 scores, death was linked more to cancer than to cardiac events. At the end of the 12-year follow-up, cancer death rates were 2.4-fold higher than those for CVD death.

Some studies attempted to assess whether there are significant associations between CAC scores and other risk factors. Knapper et al. [41] evaluated 9715 asymptomatic young subjects with and without a family history (FH) of CAD for 15 years. The results showed that the death rate varied between 4.7 and 25% in patients with FH while from 5 to 38% in those without; for this reason, they believed that the death rates in patients with FH were primarily due to age and the presence of other risk factors, suggesting that CAC score estimation is not helpful in younger subjects. Similar results were found in the study by Radford et al. [42], where they assessed whether the progression of the CAC score could change the prognostic value of the risk of cardiac events.

They underlined how the progression of the CAC score was linked to the probability of developing cardiac events but also demonstrated that the introduction of a second CAC estimation did not modify the predictive model output. Others evaluated CAC scores between hypertensive patients [43,44] or smokers [45], strengthening the added value of CAC score (≥400) compared to traditional risk factors alone for risk assessment and mortality.

Other studies evaluated whether there were any differences between men and women or between race/ethnicity in the CAC score and, consequently, in the definition of risk. Orimoloye et al. [17] demonstrated in a large asymptomatic cohort that subjects most at risk were black women due to the high prevalence of calcifications and whites and Hispanic men compared to the complementary genders of the other race groups. Kelkar et al. [46] assessed whether there were differences between women and men at 14.6 years of follow-up. The results showed that women had a 5% mortality rate with CAC = 0 and a 23.5% for those with CAC ≥ 400, compared to 3.5% and 18.0% for men, respectively. Those at risk were women aged ≥ 55 years and with CAC > 10 compared to men. Finally, some studies evaluate aspects related to the CAC score for prognostic purposes. Lathi et al. [47] assessed if left main (LM) CAC predicted mortality in 28,147 asymptomatic patients with CAC score > 0 after 12 years. LM CAC was present in 21.7% of patients and was associated with an increased hazard for all-cause death (HR 1,2 [1.1–1.3]) and CVD death (HR 1.3 [1.1–1.5]) concerning the total CAC score and risk factors. For this reason, they believed that when present, LM CAC should be reported, given the 20–30% association between developing cardiovascular and total mortality. Cho et al. [48], in their study, assessed whether CTCA could add value to CAC score-based models for prognostic mortality risk. The results found that CTCA helped assess the degree of stenosis and plaque composition by adding prognostic value to models only on the basis of risk factors but not on the basis of CAC score, which remains better for prognostic purposes in asymptomatic patients. Dzaye et al. [49] evaluated CAC-DRS in asymptomatic patients for prognostic purposes of developing CVD and all-cause mortality. The results found that patients belonging to the high CAC-DRS group had a higher risk than those with CAC-DRS A0. For this reason, they supported the idea that this system was better than just the CAC score for prognostic purposes, and therefore they suggested it to the new SCCT guidelines. See Table 1 for more details for patients characteristics.

#### 3.3.2. CAC Score in Diabetic Patients

Diabetic patients had a higher risk of developing cardiovascular disease leading to a cardiac event, e.g., heart attack or, most importantly, death. Valenti et al. [50] evaluated the prognostic utility of CAC score in 9715 diabetic patients asymptomatic versus non-diabetic for 15 years. The rate of incident mortality was higher between diabetic versus nondiabetic patients (23.2% vs. 8.4%, respectively (*p* < 0.001)). Diabetic patients with CAC = 0 had a similar prognosis to non-diabetic patients with a 5-year follow-up, whereas if we considered the presence of the CAC score associated with diabetes at a 15-year follow-up, we had an almost 2.5-fold risk of death and the worst prognosis. They underlined how important the follow-up time was, as too long a time could lead to a progression of the CAC score also on the basis of the patient’s lifestyle and, at the same time, change the risk assessment. In another study [9] on 85 diabetic patients followed for 48 months, the authors showed how patients with CAC = 0 had an excellent prognosis. They found that CAC score values > 86.6 were an independent predictor of cardiac events in 80% of patients with a specificity of 74.7%. Patients who exceeded that cutoff had an increased risk of 10.7 times higher.

Malik et al. [51] used the CAC score as a prognostic tool to evaluate the incidence of coronary heart disease (CHD) and atherosclerotic cardiovascular disease (ASCVD) among diabetes, metabolic syndrome (MetS), or neither condition patients. The statistical analysis showed that the CAC score was better for stratification and reclassification of the risk of patients with MetS and diabetes (NRI of 0.22 in the MetS group and 0.25 in the diabetes group) compared to the global risk assessment using the Framingham risk score (FRS) or ASCVD pooled cohort risk score. Their results supported the idea that the CAC score was more related to the risk of ASCVD than how long the patient has had diabetes. CAC = 0 identified patients with a low risk of having a cardiac event, regardless of diabetes duration, insulin use, or glycemic control.

Other studies have tried to evaluate prognostic differences between diabetic men and women since it was assumed that diabetic women were at increased risk of developing cardiovascular disease. Shaikh et al. [52] studied a court of 25,663 patients with and without diabetes. During follow-up of 22 years, the all-cause mortality rate was low in patients with CAC = 0 (2.6% and 3.9% in female and males) diabetes patients, while patients with CAC score > 300 had an almost six- and threefold increased risk of mortality with respect to people with no or low CAC score. This study suggested that females with diabetes had the highest risk of long-term mortality with an increasing CAC score compared to males. The presence of coronary calcifications indicated a worse prognosis for females with diabetes. Palmieri et al. [53] also maintained that the CAC score was an excellent prognostic tool; the cardiovascular risk calculated at 10 years with the CAC score was 10–20% in 78% of diabetic patients and 28% in non-diabetics, while >20% in 11% of diabetics but not in non-diabetics. See Table 2 for more details on diabetic patients.

#### 3.3.3. CAC Score Associated with SPECT

Some studies have evaluated whether combining the CAC score with the single-photon emission computed tomography (SPECT) was possible to obtain a better prognosis for patients and, at the same time, whether the information was possible was complementary or different between them. Huang et al. [54] assessed the diagnostic power of the CAC score associated with the information obtained with SPECT to evaluate the long-term (28.4 ± 9.1 years) occurrence in the Chinese population of major adverse cardiac events (MACEs). The statistical results of 1876 patients found that increasing the CAC score value led to a significant association with a high frequency of an abnormal SPECT (all *p* < 0.05) and, consequently, a higher incidence of MACEs in this type of patients compared to those with a normal SPECT and a severe or moderate CAC score. According to them, the information obtained from the two methods was independent and complimentary for a better prognosis of MACEs; simultaneously, they suggested adding the information of the CAC score to the SPECT in asymptomatic patients with suspected CAD better defines the risk. Chang et al. [55] tried to evaluate the prognostic value of the CAC score, ETT exercise treadmill testing, and SPECT in 1175 patients, primarily asymptomatic, followed for approximately 6.9 years. Statistical analyses found that patients with a cardiac event had a high CAC score, often ETT ischemia, or an abnormal SPECT.

CAC score was always stronger prognostically than other methods; in fact, there was a reclassification of 50.7% of patients, and the global value of chi-squared went from 11.72 to 45.33 (*p* < 0.0001). The clear superiority of the CAC scores over ETT and SPECT information to identify low- and high-risk patients where functional testing was more indicated was shown; on the contrary, the CAC score could be the first to be performed. The presence of CAD was familiar among patients with end-stage renal disease (ESRD). SPECT also had a low sensitivity, so Havel et al. [56] evaluated the information of the CAC score associated with that of SPECT in 77 ESRD individuals followed for about 26.4 months. A significant association between high CAC score values and severe perfusion abnormality was found in patients who had a cardiac event (CE) (*p* < 0.001, *p* = 0.0056, respectively), while patients with normal SPECT and high CAC scores did not have CE. They supported the idea that the information obtained by the two methods was independent and helpful in defining the risk; simultaneously, the CAC score could be helpful when subjects with false-negative SPECT studies with normal perfusion and with CAC = 0 was the best prognosis for the patient. See Table 3 for more details.

#### 3.3.4. CAC Score Associated with Biomarkers, Imaging, and Clinical Parameters

Over the years, many studies have strengthened the prognostic value of the CAC score in asymptomatic patients; others have tried to evaluate whether there were biomarkers, imaging parameters that could be associated with the information of the CAC score or that could be independent and provide a prognostic estimate similar to that of the CAC score. Serra et al. [57] evaluated the use of CTCA and the CAC score for the screening for atherosclerosis compared to the systematic coronary risk evaluation (SCORE) algorithm prognostics of cardiovascular events in 226 patients followed for more than 10 years. Using only the score algorithm, CTA, and the CAC score allowed for a better prognosis for patients defined as intermediate or high risk. The 10-year risk of a cardiac event was significant for patients who experienced atheroma using CTA and CAC scores.

Von Scholten et al. [20] evaluated the relationship between N-terminal pro-brain natriuretic peptide (NT-proBNP) and CAC score for prognostic purposes in 200 asymptomatic patients with type 2 diabetes and microalbuminuria (considered to be those at highest risk of cardiovascular disease) followed for 6.1 years. Patients with NT-proBNP > 45.2 ng / L or CAC ≥ 400 were at high risk with NT-proBNP < 45.2 ng/L and low-risk CAC < 400. Higher NT-proBNP values were strongly predictive and associated with high CAC scores of CVD, where high NT-proBNP was associated with depressed systolic and diastolic function, while CAC correlated with atheromatous plaque formation. Dikic et al. [58] found the association between coronary flow velocity reserve (CFRV) and CAC score in 200 asymptomatic and diabetic patients. In diabetic patients with a high CAC score, there was a significant correlation between CFRV and total CS compared to non-diabetics, where there was no such correlation. Patients with CS > 200 and CFRV < 2 had the worst prognosis and, therefore, the highest risk of developing cardiac events within one year, approximately 24.3-fold compared to patients with CS < 200 and CFRV ≥ 2. They suggested that the two parameters provided helpful information if analyzed alone, but if they were evaluated together, they were complementary, thus obtaining a better prognosis. Some studies believe that the CAC score was of little use compared to the Framingham risk score (FRS) and the degree of stenosis. In contrast, others report better results where the CAC score was superior to conventional biomarkers. Park et al. [59] evaluated the predictive power of clinical parameters, biomarkers, and imaging parameters for the development of cardiovascular diseases in about 5182 asymptomatic patients followed for 48 months. They used four regression models to evaluate the predictive power of each parameter as the FRS, CACs, the degree of stenosis, and the value of high-sensitivity C-reactive protein (hsCRP). Patients with FRS ≥ 15% or more significant stenosis had the worst prognosis when evaluating survival curves. The degree of coronary artery stenosis calculated with CT and FRS was an independent prognostic tool in asymptomatic patients. The analysis of multiple regression models found that the CACs alone was an excellent prognostic tool. However, when added to regression models with other parameters, the degree of stenosis performed better than CACs as a prognostic tool. The opposite results seemed to be found in the study by Rana et al. [60], which evaluated the predictive value of the CAC scores concerning different biomarkers for prognostic purposes of developing CVD in 1286 asymptomatic patients followed for 4.1 ± 0.4 years.

The biomarkers studied were C-reactive protein, interleukin-6, myeloperoxidase, B-type natriuretic peptide, and plasminogen. Each biomarker was evaluated in predictive models for CVD events with adjustment for risk factors. The CAC score increased the c-statistic and efficiently redefined patient risk when added to statistical models. The predictive power of CACs compared to biomarkers was evident since the latter could not redefine or improve the definition of patients’ disease risk. Not only imaging parameters or biomarkers were associated with the CAC score, but they also exercised, as Choi et al. [61] point out in their study, a significant difference in the effect of the CAC score on all-cause mortality in patients with low or high exercise capacity. The results showed that in subjects with high exercise capacity, high CACS values had little influence on all-cause mortality compared to those with lower exercise capacity. See Table 4 for more details.

#### 3.3.5. Coronary Computed Tomography Angiography (CCTA) vs. CAC Score

Eleven studies evaluated CCTA as a better prognostic tool than the CAC score in defining the risk of developing cardiovascular disease. Among them, six believed that CCTA is higher than the CAC score, and three evaluated the predictive power of CCTA in diabetic patients—one in patients with stroke, and another in elderly patients. Moon et al. [62] found by dividing 470 asymptomatic patients into groups with regular, obstructive, and nonobstructive CAD that the highest percentage of patients with CAC > 400 were present in the obstructive CAD group. Survival also differed significantly; in fact, the percentage of cardiac events decreased among the various groups on the basis of the number of vessels involved. It underlined how CCTA, if added to the predictive model on the basis of FRS or CACS, provided a better risk estimate with an increased C-index (from 0.698 to 0.749) and increased category-free net reclassification index (0.478; *p* = 0.022). CAC score was useful but did not allow for the direct study of the state of health of the coronary arteries, any stenosis, or the characterization of the plaques. These factors could affect the estimation of risk. Whether one of these factors could affect the subjects’ prognosis is found in the work of Takamura et al. [63]. In their study, they found that CT verified high-risk plaque (CT –HRP) was an independent predictor (HR 11.27, *p* < 0.0321) and could be used to improve risk estimation on the basis of CAC score information in asymptomatic patients. Plank et al. [64] found that CAC = 0 did not exclude the presence of non-calcified plaques in asymptomatic patients at high risk of CAD; on the contrary, in CCTA, with a direct evaluation of coronary arteries, the presence of stenosis or plaques could do. CAC score had a low predictive power (C = 0.64; 95% CI 0.558 to 0.711) versus CCTA (C = 0.71; 95% CI 0.632 to 0.77, *p* < 0.001) in the presence of stenosis. Similar results found in Dedic et al. [25] evaluated CCTA in patients considered at high risk. For patients with a CAC score = 0, there was no prognostic difference between CACS and CCTA. However, in patients with CAC between 1 and 100 or >400, CCTA was found to provide an increased C-statistic from 0.81 to 0.84 with a total net reclassification index of 0.19. To compare predictive models on the basis of CACS and CCTA, interesting results were found in the study of Cho et al. [65]. For patients with CAC < 100, or between 101 and 400, CCTA showed a marked improvement in the risk estimate with NRI = 0.75 (95% CI; 0.23–1.38, *p* = 0.008) and ΔC-statistic = 0.13 (95% CI 0.03–0.23; *p* = 0.011). The surprising thing was that the predictive value for patients with a CAC score > 400 weakened, suggested for technical problems due to the acquisition (artifacts on the images); for these reasons, CCTA turned out to be more useful in intermediate-risk patients than to be used in patients with low or very high CAC score. Moreover, Yoo et al. [66] found that the detection rate of non-calcified plaques (NCP) with CCTA was higher in the low-CAC group than in the CAC = 0 group (31.5% vs. 6.9%, *p* < 0.001), and the same was the case for significant strictures (7.5% vs. 0.8%, *p* < 0.001). They claimed that NCPs had predictive factors such as diabetes, hypertension, and LDL-cholesterol, but at the same time, patients with CAC = 0 shown to have subclinical atherosclerosis and significant stenosis, and for these reasons they suggested the use of CCTA in patients with risk factors and with low CACS. In diabetic patients, Hoogen et al. [67] found that 85% had CAD and 51% had non-obstructive CAD; particularly obstructive (50–70%) or severe CAD (>70%) was predictive of cardiac events (HR 11.10 and HR 15.16, respectively; *p* = 0.001). By adding CCTA information to models only on the basis of CAC scores in diabetics, the definition of the risk of cardiac events improved, as also highlighted by Halon et al. [68] in his study. Min et al. [69] found in 64% of patients with CACS > 0 that obstructive CAD was present in 15.6% for those with CACS 1–10, 38.4% those with CACS 101–400, and 64.3% those with CACS 400. The results suggested that increased cardiac events were associated with the number of obstructive vessels, segment stenosis, and maximal stenosis. However, it was evident that CCTA improved risk assessment, stratification, and reclassification in diabetic patients. Even in patients with stroke without chest pain in the study by Hur et al. [70], the results suggested the usefulness of the CCTA compared to models only on the basis of CACs (iAUC: 0.863 vs. 0.752) in assessing the risk of developing MACEs. One of the critical factors that can contribute to the development of cardiac events is the age of the patients. Han et al. [24], in their study, found that when CCTA information was added to the predictive models in young patients, there was no added value compared to CACs + FRS, while the difference was in elderly patients with an increase in the C-statistic (0.75 vs. 0.70, *p* = 0.015). For this reason, they suggested the use of CCTA in older patients and not in young people. See Table 5 for more details.

#### 3.3.6. Quality Assessment

Results of the QUIPS assessment are shown in Figure 2 and reported in Appendix A. The risk of bias was ranked low or moderate across all studies for all the six QUIPS domains. Most studies displayed a low risk of bias in the domains of study, attrition, outcome measurement, study confounding, and statistical analysis and reporting. All studies were judged to be a low risk of bias for prognostic factor measurement. In contrast, there was a higher percentage of studies with a moderate risk of bias concerning the study participation domain.

## 4. Discussion

In this systematic review, we aimed to investigate the role of the CAC score as a prognostic tool to predict the presence of CAD in asymptomatic patients. In the last decade, attention to the use of the CAC score has shifted from symptomatic to asymptomatic patients, in which preventive therapies such as the use of statins [3] or correct management of risk factors (smoking, diabetes, hypercholesterolemia, and others) are very effective and can prevent the onset of cardiac events even in the long term. CAC score is an easy-to-perform, reproducible, and reliable test for estimating CAD risk. In this scenario, our systematic review can provide critical new insights and help to reach a standard view on the use of CAC score for CAD prognosis. After inclusion and exclusion criteria, we examined 45 studies from 2010 onwards, evaluating the CAC score as a prognostic tool of CAD in asymptomatic patients. The main findings and conclusions of the selected studies varied from each other; most of them agree that CAC was an excellent tool. In contrast, others believe that it was not always adequate [62,63,64,65,66,67,68] or evaluated if it was associated with other imaging methods or biomarkers [20,58,60] could suggest an inaccurate prognosis. Most of the studies indicated a strong correlation between the increase in the CAC score and the occurrence of cardiac events; in particular, the patients most at risk were those with CAC 400–1000 or >1000. In some cases, even those who had CAC > 100 associated with risk factors had a higher risk than subjects with CAC = 0 [31,34,35,36,37]. The latter had the best prognosis; they were more likely to die from other causes such as cancer than from CAD or cardiac events [40]. Many studies have tried to evaluate which factors could be related to the increase in the CAC score, such as hypertension, smoking, familiarity with CAD, and age [31,33,41,43,44,45]. However, all have one aspect in common that was they underlined the reliable prognostic power of the CAC score, regardless of the risk factor associated. Even in diabetic patients, the CAC score is a very reliable tool for the prognostic purposes of CAD; the best prognosis for diabetic patients was to have a low CAC or equal to 0 compared to non-diabetic and diabetic with CAC > 0 [9,50,51,52,53]. Regarding the CAC score as an excellent prognostic tool, many studies tried to verify some problems such as if there was a threshold value beyond which it was sure in terms of the development of cardiac events, what the warranty period of the subjects with CAC = 0 was, and whether there were prognostic differences with different CAC scores between men and women or between races/ethnicities. There was no specific threshold value beyond which a cardiac event occurs. However, it was clear how a subject with CAC = 0 at 10 years of follow-up had a 90% probability of survival and did not develop CAD compared to subjects with CAC > 200 who at 10 years had a 50% chance of surviving [37]. The warranty period of CAC = 0 was defined as the period within which the subject’s risk did not change while remaining low. Some studies had estimated that the warranty period for subjects with CAC = 0 was about 15 years, within which no cardiac events occurred [38,39,40]. The results underline that if the subjects did not change their lifestyle excessively, favoring, in that case, the progression of the CAC score and the change in risk, the prognosis also remains the same after years. The CAC score may be considered appropriate to repeat by the clinician to improve risk estimation, but for prognostic purposes, it does not change much [42]. On the contrary, others argue that the appropriate period to repeat the CAC score is about 3–4 years, as they believe that the score changes and, at the same time, the prognosis [11,12]. Interesting results emerged when CAC scores were evaluated in the different races/ethnic groups. Some obtained results consistent with the literature CAC = 0 were reliable in diagnosing CAD in asymptomatic Korean and Hispanic populations [13,14]. Different CAC score values were found concerning the average Western CAC score values; in Arab women, the average values were higher than in Western women [15]. On the contrary, in Japanese men, the values were lower than in Westerners [16]. Orimoloye et al. [17] in their study included 38,277 whites, 1621 Asians, 977 blacks, and 1349 Hispanics followed for 11.7 years. The patients most at risk were blacks and Hispanics as opposed to Asians and whites. The differences in risk estimation between races/ethnicities could be associated with cultural, social, or economic factors. Future studies could investigate if factors such as eating habits affect the risk of developing CAD. There was a prognostic difference between women and men. The mortality rate in women with CAC = 0 was 5%, while it was 23.4% with CAC > 400; in men, 3.5% for those with a CAC = 0 and 18.0% for those with a CAC score ≥ 400. Women aged > 55 and with CAC > 0 were highest risk. By comparing the association between SPECT and CAC score for prognostic purposes, the results showed that they were two useful independent tools whose predictive power increased when associated; in fact, often, increased CAC score values were associated with abnormal SPECT results. Unlike the SPECT, the CAC score was more reliable as there was no risk of a false positive resulting in an over or underestimation of the risk [54,55,56]. There are some limitations related to the CAC score as a prognostic tool. Many studies have evaluated the association of the CAC score and CTA for prognostic purposes, highlighting that CAC is a good and reliable tool; however, in some cases, it could not be used alone. CTA allows for the direct visualization of the state of the coronary arteries and the characterization of the plaques. Some studies show that patients with CAC = 0 are less likely to develop CE, but this does not mean they do not have non-calcified plaques or strictures that could create heart problems [64,66]. CTCA improves the prognosis for patients at intermediate risk with CAC between 1 and 100 or >400 [25]; in fact, many increase the C-statistic when adding CTCA in predictive models [62,63,65,68,69,70]. Another disadvantage that could hinder using the CAC score in asymptomatic patients, perhaps even young ones, is the radiation dose provided during the exam. Many studies have evaluated possible solutions to solve the problem as modifying the acquisition kilovolts passing from 120 to 70–80 Kv [26] or using adequate protocols with tin filter and IBHC calcium material reconstruction [27], which allowed for a dose reduction of 80%. Future developments of the CAC score could be the application of machine learning and deep learning algorithms to detect CAC and prognosticate the risk of CAD in a more precise and faster way. The future prospective is to use the CAC score as a prognostic tool in young and old patients, asymptomatic and perhaps with risk factors, while avoiding CTCA, is helpful in some cases. Future studies should investigate the related question in what circumstances one tool is better than the other and when it is necessary to use the CTCA together. There are also other aspects of the CAC score to be evaluated even more precisely, such as the warranty period of the CAC = 0 or how the score differences between races/ethnicities were created, and therefore further studies on larger populations are needed.

### Study Limitations

Our study is not without limitations. First, 15 of the selected studies [25,31,34,36,39,40,44,45,47,49,54,59,61,63,66] used a retrospective design. Second, some studies showed results that conflicted with the main notion of the prognostic value of CAC scores [24,62,63,64,65,67,68]. Due to the different equipment and imaging protocols for calculating the CAC score [41,49,51,52,53]; the heterogeneity of the participant population [17,33]; and some studies having a small sample, including [20,53,56,57,58,67], the results are show difficulty in leading back to a common organic concept. In order to make our study uniform, since many studies had different purposes, we only evaluated the prognostic aspect of the CAC score in asymptomatic individuals. Thus, our conclusions based on findings related to this aspect are solid.

## 5. Conclusions

In conclusion, the CAC score is a consolidated, reliable, repeatable, and accurate prognostic tool for estimating the risk of developing CAD, especially in asymptomatic patients with risk factors. The CAC score allows for acting in the field of primary prevention thanks to the correct estimation of risk, providing the patient with personalized therapy aimed at controlling or eliminating risk factors while providing a low risk of developing cardiac events or CAD years later.

## Figures and Tables

**Figure 1 jcm-11-05842-f001:**
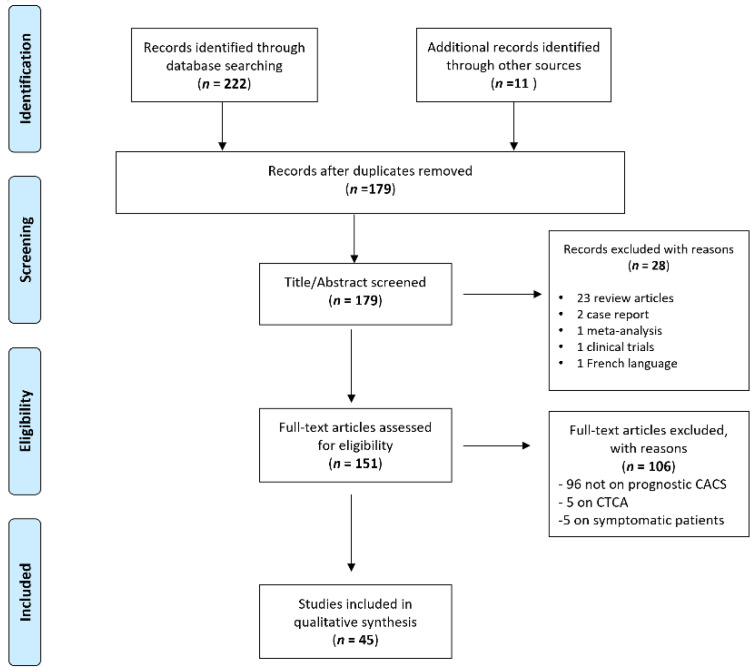
Preferred Reporting Items for Systematic Reviews and Meta-Analyses (PRISMA) flow diagram.

**Figure 2 jcm-11-05842-f002:**
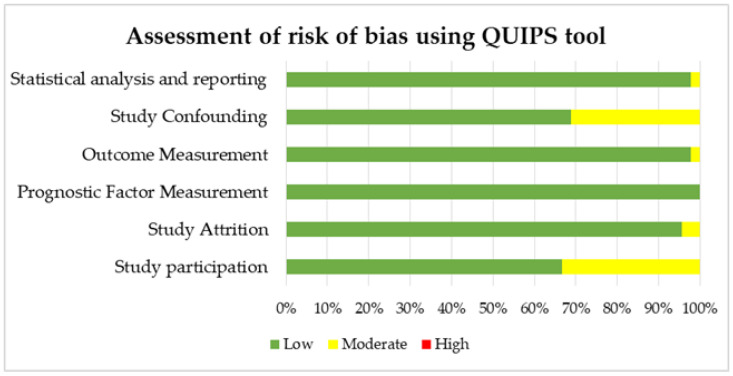
Risk of bias assessment according to the six domains of the Quality in Prognostic Studies (QUIPS) tool for the 45 studies included in the systematic review.

**Table 1 jcm-11-05842-t001:** The most important parameters of each article included in this systematic review.

Author, Year	N. of Patients	Mean Age (Years)	Study Design (P/R)	Pts	Risk Factors	Imaging	Scan Details	Follow-Up (Years)	Statistical Analysis	Main Findings
Dzaye et al. [49], 2020	54,678	54.2	R	ASX	DLP, HT, DM, SS, FH of CAD	EBCT, CT	MDCT manifacturer NR	11.7	CPhM AUC analysis	The CAC-DRS system, combining the Agatston score and the number of vessels with CAC provides better stratification of risk for CHD, CVD, and all-cause death than the Agatston score alone.
Blaha et al. [40], 2020	66,363	54.5	R	ASX	DLP, HT, DM, SS, FH of CAD	EBCT, CT	NR	12	Multivariate regression CPhM	CAC = 0 represents a unique population with favorable all-cause prognosis. Detection of any CAC in young adults could be used to trigger aggressive preventive interventions.
Dudum et al [31], 2019	14,169	48.1	R	ASX	DLP, HT, DM, SS	CT	C-speed scanner GE Imatron, 4 slice MDCT scanner Siemens, GE 64 slice lightspeed	11.6	KMA, unvariate CPhM, ROC curves	CAC scoring was a reliable predictor of all-cause, CVD, and CHD mortality.
Lahti et al. [47], 2019	28,147	58.3	R	ASX	DLP, HT, DM, SS, FH of CAD, BMI	CT, EBCT	64 slice GE	NR	Regression CPhM	The presence and high burden of left main CAC are independently associated with a 20–30% greater hazard for cardiovascular and total mortality in asymptomatic adults.
Orimoloye et al. [17], 2018	42,224	54.7	P	ASX	DLP, HT, DM, SS, FH of CAD	EBCT, CT	NR	11.7	KMA, multivariate CPhM, Fine and Gray proportional subhazards model	CAC predicts all-cause and CVD mortality in all studied race/ethnicity groups, including Asians and Hispanics, who may be poorly represented by the Pooled Cohort Equations.
Cho et al. [48], 2018	1226	58	P	ASX	DLP, HT, DM, SS, FH of CAD	CT	CT 64 slices or greater	5.9 ± 1.2	KMA, regression CPhM	CCTA does not offer added value when CCTA findings were added to model RF + CACS at 6 years of follow-up.
Cho et al. [32], 2017	6656	59	P	NR	NR	CT	MDCT 64 rows or greater	5.1	KMA, uvariate and multivariate CPhM	Patients with CAC score ≥100 and no coronary luminal narrowing experience death rates similar to those with non-obstructive CAD.
Carr et al. [33], 2017	5115	32 to 56	P	NR	DLP, HT, DM, SS, FH of CAD, BMI	CT	NR	12.5	Regression CPhM	The presence of CAC among individuals aged between 32 and 46 years was associated with increased risk of fatal and nonfatal CHD during 12.5 years of follow-up.
Radford et al. [42], 2016	5933	49.2	P	NR	DLP, HT, DM, SS, FH of CAD, CFR	EBCT	C-150XP or C-300 models (Siemens); slices of 3 mm thickness were obtained with 2 mm table increments	7.3	The Ionckheere-Terpstra nonparametric method; the Wald method, CPhM, Harrell’s c-index	If serial CAC scanning is performed, the latest scan should be used for risk assessment, and in this context, CAC progression provides no additional prognostic information.
Lee et al. [39], 2016	48,215	54.1	R	ASX	DLP, HT, DM, SS, FH of CAD	CT	Philips Brilliance 256 iCT, Philips Brilliance 40 channel MDCT, Siemens 16-slice Sensation and GE 64-slice Lightspeed,225–400-ms gantry rotation time	4.4	KMA, unvariate and multivariate regression CPhM	In asymptomatic Korean adults, the absence of CAC evoked a strong protective effect against ACM as reflected by longer warranty period, when no other RF were present.
Kelkar et al. [46], 2016	2363	55.0	P	ASX	DLP, HT, DM, SS, FH of CAD	EBCT, CT	NR	14.6	Unvariate and multivariate regression CPhM, Harrel C-statistic	CAC effectively identifies high-risk women with a low-intermediate risk factor burden.
Knapper et al. [41], 2016	9715	40 to 70	P	ASX	DLP, HT, DM, SS	EBCT, CT	NR	14.6	Unvariate and multivariate regression CPhM	For younger and lower-risk FH cohorts, CAC screening did not provide additive prognostic information beyond that of the traditional cardiac risk factors.
Han et al. [34], 2015	34,386	53.8	R	NR	DLP, HT, DM, SS, FH of CAD, BMI	CT	Philips Brilliance 256 iCT, Philips Brilliance 40 channel multi-detector CT, Siemens 16-slice Sensation, and GE 64-slice Lightspeed	4.9	KMA, regression CPhM	In an asymptomatic Korean population, CACS improved prediction of all-cause mortality over and above that of a conventional risk tool.
Valenti et al. [38], 2015	9715	53.4	P	ASX	DLP, HT, DM, SS, FH of CAD, BMI	EBCT	C-100 or C-150 Ultrafast CT GE Imatron, slice thickness of 3 mm, slices = 40, using a 100 ms/slice scanning time	14.6	Mann–Whitney test, multivariate regression CPhM, AUC analysis	In individuals considered at high risk by clinical risk scores, a CAC score of O confers better survival than in individuals at low-to-intermediate risk but with any CAC score.
Shaw et al. [36], 2015	9715	40 to 80	R	ASX	DLP, HT, DM, SS, FH of CAD	EBCT, CT	NR	14.6	Univariable and multivariable Cox regression model, Hosmer–Lemeshow test	The extent of CAC accurately predicts 15-year mortality in a large cohort of asymptomatic patients.
Patel et al. [37], 2014	44,052	60	P	ASX	DLP, HT, DM, SS, FH of CAD, BMI	EBCT	C-100 or C-150 Ultrafast CT GE Imatron, slice thickness of 3 mm, slices= 40, using a 100 ms/slice scanning time	5.6 ± 2.6	KMA CPhRM	Increasing calcified plaque in coronary arteries continued to predict a graded decrease in survival among patients with extensive Agatston score > 1000 with no apparent upper threshold.
Graham et al. [43], 2012	44,052	55	P	ASX	DLP, HT, DM, SS, FH of CAD, BMI	EBCT	NR	5.6 ± 2.6	KMA, regression CPhM	Addition of CAC scores contributed significantly to predicting mortality in addition to only traditional risk factors alone among those with and without hypertension.
Mcevoy et al. [45], 2012	44,042	54	R	ASX	DLP, HT, DM, SS, FH of CAD, BMI	EBCT	C-100 or 150 Ultrafast CT GE	5.6 ± 2.6	KMA, regression CPhM	Smokers with any CAC had significantly higher mortality than smokers without CAC.
Nasir et al. [35], 2012	44,052	54	P	ASX	DLP, HT, DM, SS, FH of CAD, BMI	EBCT	C-100 or a C-150 Ultrafast CT GE, slice thickness of 3 mm, slices = 40, 100 ms/slice scanning time	5.6 ± 2.6	KMA, regression CPhM	Individuals without RFs but elevated CAC have a substantially higher event rates than those who have multiple RFs but no CAC; these findings challenge the exclusive use of traditional risk assessment algorithms for guiding the intensity of primary prevention therapies.
Shemesh et al. [44], 2011	423	64	R	ASX	DLP, HT, DM, SS, FH of CAD, left ventricular hypertrophy	CT	Dual detector spiral CT without electrocardiogram gating	14 ± 0.5	Mann–Whitney test, CPhM, C-index	CAC is associated with long-term mortality in asymptomatic hypertensive adults.

Abbreviations: ASX = asymptomatic, AUC = area under the curve analysis, CACS = coronary artery calcium score; CT = computed tomography; NRI = net reclassification index; CE = cardiac events; CPhM = Cox proportional hazard model analysis, EBCT = electron beam computed tomography; ETT = excise tolerance test; DLP = dyslipidemia, DM = diabetes mellitus; FH = family history; HT = hypertension; HCL = hypercholestrerolemia; KMA = Kaplan–Meier analysis; KMSA = Kaplan–Meier survival analysis; NR = not reported; P = prospective, R = retrospective, ROC = receiver operating characteristic curve; SPECT = single-photon emission computed tomography; SS = smoking status, SX = symptomatic; N. = number; Pts. = patients.

**Table 2 jcm-11-05842-t002:** The most important parameters on diabetic patients.

Author, Year	N. of Patients	Mean Age (Years)	Study Design (P/R)	Pts	Risk Factors	Imaging	Scan Details	Follow-Up (Years)	Statistical Analysis	Main Findings
Shaik et al. [52], 2019	25,663	55.27	P	ASX DM, not DM	DLP, HT, DM, SS, FH of CAD	EBCT, CT	C-150 XL Ultrafast CT GE, MDCT 64 slice lightspeed GE, FOV = 35 cm, matrix size = 512 × 512,120 kVp, slice thickness = 3 mm.	14.7 ± 3.8	KMA, unvariate and multivariate CPhM	The absence of CAC was associated with very low cardiovascular as well as all-cause mortality events in all subgroups during long-term follow-up.
Malik et al. [51], 2017	6814	62.2	P	MetS and diabetes	DLP, HT, DM	EBCT, CT	NR	Follow-up, extended to the first occurrence of CE	KMA, regression CPhM	The addition of CAC score to global risk assessment was associated with significantly improved risk classification in those with MetS and diabetes.
Palmieri et al. [53], 2017	38	64	P	ASX	DLP, HT, DM, SS, FH of CAD	CT	Aquilion 64 multislice scanner Toshiba, slice thickness 0.5 mm, 120 kV and 300–450 mA	180 days	Chisquare and Fisher’s exact test	On the basis of CAC, in the presence of non-obstructive carotid atherosclerosis, asymptomatic DM may show significantly higher CAD burden than non-DM, even in the absence of inducible myocardial ischemia.
Valenti et al. [50], 2016	9715	53.4	P	ASX DM and not DM	DLP, HT, DM, SS, FH of CAD	EBCT	C-100 or C-150 Ultrafast CT GE Imatron, slice thickness = 3 mm, slices = 40, using a 100 ms/slice scanning time	15	Mann–Whitney test, multivariate regression CPhM, KMA	CAC = 0 is associated with a favorable 5-year prognosis for asymptomatic diabetic and nondiabetic individuals.
Faustino et al. [9], 2014	85	60	P	ASX DM type 2	DLP, HT, DM, SS, FH of CAD	CT	CACS: 8 × 3 mm collimation, 55 mAs, 120 kV, 3 mm width. CTA: 16 × 0.75 mm collimation, 400 ms gantry rotation, pitch = 0.298, 120 kV, 600–800 mAs	48 months	Cox regression (method forward conditional), ROC curve, AUC analysis	CS showed great value in T2DP risk stratification, and its prognostic value was further enhanced by CTA data.

Abbreviations: ASX = asymptomatic, AUC = area under the curve analysis, CACS = coronary artery calcium score; CT = computed tomography; NRI = net reclassification index; CE = cardiac events; CPhM = Cox proportional hazard model analysis, EBCT = electron beam computed tomography; ETT = excise tolerance test; DLP = dyslipidemia, DM = diabetes mellitus; FH = family history; HT = hypertension; HCL = hypercholestrerolemia; KMA = Kaplan–Meier analysis; KMSA = Kaplan–Meier survival analysis; NR = not reported; P = prospective, R= retrospective, ROC = receiver operating characteristic curve; SPECT = single-photon emission computed tomography; SS = smoking status, SX = symptomatic; N. = number; Pts. = patients.

**Table 3 jcm-11-05842-t003:** The most important parameters on CAC score associated with SPECT.

Author, Year	N. of Patients	Mean Age (Years)	Study Design (P/R)	Pts	Risk Factors	Imaging	Scan Details	Follow-Up (Years)	Statistical Analysis	Main Findings
Huang et al. [54], 2019	1876	58.0	R	ASX	DLP, HT, DM, SS, FH of CAD, BMI	CT, SPECT	CT: High-Definition XT GE, 40/48 slices, 2.5 mm section thickness; 120 kV,125 mA; SPECT: triple-head camera using a low-energy, high-resolution, parallel-hole collimator with a rotation in a continuous mode	28.4 ± 9.1	KMSA, regression CPhM	The authors support adding a CACS testing in addition to SPECT in asymptomatic patients to better define the risk of cardiac events during follow-up.
Chang et al. [55], 2015	988	57.5	P	ASX or SX	DLP, HT, DM, SS, FH of CAD	EBCT, ETT, SPECT	Imatron C-150	6.9	KMA, unvariate CPhM, AUC, global chi-squared	CACS as a first-line test over ETT or SPECT for accurability assessing long-term risk in such patients.
Havel et al. [56], 2015	77	59.5	P	35 DM patients; there was a history of previous MI in 6 patients	NR	SPECT, CACS FROM PET/TC	PET/TC Biograph 16 Siemens	26.4 months	KMA, CPhM	This study suggests that combined evaluation of MPI and CAC can predict the outcome in ESRD individuals, while severe perfusion abnormality on gated-SPET and high CAC score ≥ 1000 are predictors of future cardiac events.

Abbreviations: ASX = asymptomatic, AUC = area under the curve analysis, CACS = coronary artery calcium score; CT = computed tomography; NRI = net reclassification index; CE = cardiac events; CPhM = Cox proportional hazard model analysis, EBCT = electron beam computed tomography; ETT = excise tolerance test; DLP = dyslipidemia, DM = diabetes mellitus; FH = family history; HT = hypertension; HCL = hypercholestrerolemia; KMA = Kaplan–Meier analysis; KMSA = Kaplan–Meier survival analysis; NR = not reported; P = prospective, R = retrospective, ROC = receiver operating characteristic curve; SPECT = single-photon emission computed tomography; SS = smoking status, SX = symptomatic; N. = number; Pts. = patients.

**Table 4 jcm-11-05842-t004:** The most important parameters on CAC score associated with biomarkers, imaging, and clinical parameters.

Author, Year	N. of Patients	Mean Age (Years)	Study Design (P/R)	Pts	Risk Factors	Imaging	Scan Details	Follow-Up (Years)	Statistical Analysis	Main Findings
Serra et al. [57], 2019	266	55.4	P	ASX	HT, HCL, DM, serum C reactive protein	CT	16-slice MDCT Philips	>10	KMA, the Mantel–Haenszel test	CTA and CCS assessments had a higher OR than that associated with assessments of patients at intermediate risk using the SCORE algorithm.
Choi et al. [61], 2016	25,972	53.7	R	ASX	DLP, HT, DM	CT	Philips brilliance 256 iCT, Philips 40 channel multidetector, Siemens 16 slice sensation, GE 64 slice lightspeed with 225–400 ms gantry rotation	5.5	CPhM	The effect of high CACS on all-cause mortality is lessened by good exercise capacity in the asymptomatic population.
Dikic et al. [58], 2015	200	57.7	P	101 ASX with DM and 99 ASX without DM	DLP, HT, DM, SS, FH of CAD	CT	Somatom Sensation 64 Siemens; 100 ms scan time, 3 mm slice tickness, 40–45 slices	1	KMA, unvariate CPhM, AUC	DM patients with CACS > 200 and CFVR < 2 had the worst outcome.
Von Sholten et al. [20], 2015	200	54 to 65	P	ASX with type 2 DM	DLP, HT, DM, SS, FH of CAD	CT	16 MDCT Philips, slice thickness 3 mm	6.4	Mann–Whitney µ test, KMA; CPhM	In patients with type 2 diabetes and microalbuminuria but without known coronary artery disease, NT-proBNP and CAC were strongly associated with fatal and nonfatal CVD, as well as with mortality.
Park et al. [59], 2013	5182	53	R	ASX	DLP, HT, DM, SS, FH of CAD, BMI	CT	64 slice Brilliance Philips, 64 × 0.625 mm section collimation, 420 ms rotation time, 120 kV, 800 mA	48 months	KMA, regression CPhM	Biomarkers and imaging parameters of cardiovascular disease, both FRS and degree of coronary artery stenosis, are independent parameters to predict adverse outcome in an asymptomatic population.
Rana et al. [60], 2012	1286	58.6	P	ASX	HT, HCL, DM, serum C-reactive protein, left ventricular disfunction and fibrinolysis	EBCT, CT	EBCT GE, MDCT Siemens	4.1 ± 0.4	Multivariate regression CPhM, Harrell c-statistic and AUC curves	Asymptomatic subjects without known CVD; addition of CAC but not biomarkers substantially improved risk reclassification for future CVD events beyond traditional risk factors.

Abbreviations: ASX = asymptomatic, AUC = area under the curve analysis, CACS = coronary artery calcium score; CT = computed tomography; NRI = net reclassification index; CE = cardiac events; CPhM = Cox proportional hazard model analysis, EBCT = electron beam computed tomography; ETT = excise tolerance test; DLP = dyslipidemia, DM = diabetes mellitus; FH = family history; HT = hypertension; HCL = hypercholestrerolemia; KMA = Kaplan–Meier analysis; KMSA = Kaplan–Meier survival analysis; NR = not reported; P = prospective, R = retrospective, ROC = receiver operating characteristic curve; SPECT = single-photon emission computed tomography; SS = smoking status, SX = symptomatic; N. = number; Pts. = patients.

**Table 5 jcm-11-05842-t005:** The most important parameters on CCTA vs. CAC score.

Author, Year	N. of Patients	Mean Age (Years)	Study Design (P/R)	Pts	Risk Factors	Imaging	Scan Details	Follow-Up (Years)	Statistical Analysis	Main Findings
Moon et al. [62], 2019	470	75.1	P	ASX	DLP, HT, DM, SS, FH of CAD, BMI, CACS	CT	64-slice MDCT Brilliance Philips	8.2	KMA, unvariate or multivariate CPhM, C-statistics, categorical and category-free NRI	CCTA showed better long-term prognostic value for MACE than coronary artery calcium score in this asymptomatic older population.
Han D. et al. [24], 2018	3145	56.6	P	ASX	DLP, HT, DM, SS, FH of CAD, BMI, CACS	CT	64-slice MDCT	26 months	Regression CPhM, Harrell’s C-index, categorical NRI	CCTA provides added prognostic value beyond cardiac risk factors and CACS for the prediction of MACE in asymptomatic older adults.
Takamura et al. [63], 2017	495	63.4	R	ASX	DLP, HT, DM, SS, FH of CAD, BMI	CT	64 slices MDCT or 320 row area detector CT (ADCT) Toshiba	716.5 ± 262.6 days	KMA, regression CPhM, ROC and AUC curves, NRI	Although the cardiac event rate was low, the evaluation of CCTA plaque characteristics may provide incremental prognostic value to CACS in asymptomatic patients.
Dedic et al. [25], 2016	665	56	P/R	ASX	DLP, HT, DM, SS, FH of CAD, BMI	CT	64-slice MDCT	3	KMA, unvariate regression CPhM, NRI	CCTA has incremental prognostic value and risk reclassification benefit beyond CACS in patients without CAD symptoms but with high risk of developing CVD.
Halon et al. [68], 2016	630	63.5	P	ASX DM type 2	DLP, HT, DM, SS, FH of CAD, BMI	CT	64 slice Brilliance CT; Philips; 120 to 140 kV, 500 to 1400 mAs, slice collimation 6490.625-mm, 0.42 s gantry rotation time, pitch 0.2 mm	6.6 ± 0.6	KMA, CPhM, ROC curves, NRI	CTA provides additional prognostic information in asymptomatic type 2 diabetics not obtainable from clinical risk assessment and CAC alone.
Van den Hoogen et al. [67], 2016	525	54	P	DM	DLP, HT, DM, SS, FH of CAD, BMI	CT	64-slice Aquillon 64 Toshiba or 320 MDCT Aquillon ONE	5	KMA, unvariate regression CPhM	Coronary CTA provided prognostic value in diabetic patients without chest pain syndrome. Most importantly, the prognosis of patients with a normal CTA was excellent.
Cho et al. [65], 2015	3217	57	P	ASX	DLP, HT, DM, SS, FH of CAD, BMI	CT	64 slice or more MDCT	2.5	KMA, continuous NRI	CCTA provides incremental prognostic utility for prediction of mortality and non-fatal myocardial infarction for asymptomatic individuals with moderately high CACS, but not for lower or higher CACS.
Hur et al. [70], 2015	350	64.1	P	Ischemic stroke patients without chest pain	DLP, HT, DM, SS, FH of CAD, BMI	CT	Somatom Definition Flash Siemens, slice thickness 3 mm, collimation 2 × 64 × 0.6 mm; gantry rotation time 280 ms; 280–380 mAs; 120 kV; pitch 0.2–0.43	409 days	KMA, regression CPhM, ROC curve method was used and the integrated area under the curve iAUC, NRI	In ischemic stroke patients without chest pain, CCTA findings of CAD provide additional risk-discrimination over CACS.
Plank et al. [64], 2014	711	54.5	P	ASX	DLP, HT, DM, SS, FH of CAD, BMI	CT	CACS: 64-slice CT Somatom Sensation Siemens; collimation 64 × 1.5 mm, 120 kV, ECG-gating, slice thickness 3 mm filter kernel B 35, CCTA: 128 Somatom Definition Flash, Siemens, Somatom Sensation 64, Siemens, collimation 2 × 64 × 0.6 mm with a z-flying spot and 64 × 0.6 mm, rotation time 0.28 and 0.33 s	2.65	KMA, CPhM, ROC analysis	CAD prevalence by CTA in asymptomatic high-risk patients is high. CCS zero does not exclude CAD. CTA is highly accurate in excluding CAD.
Min et al. [69], 2014	400	60,4	P	ASX DM	DLP, HT, DM, SS, FH of CAD, BMI	CT	64-slice CT	2.4 ± 1.1	Cox regression analysis	For asymptomatic diabetic individuals, CCTA measures of CAD severity confer incremental risk prediction, discrimination, and reclassification on a per-patient, per-vessel, and per-segment basis.
Yoo et al. [66], 2011	7515	50.1	R	ASX	DLP, HT, DM, SS, FH of CAD, BMI	CT	64-slice MDCT Brilliance 64 CACS: 120-Kv, 55 mAs, 2,5 mm scan thickness, CCTA: 64 × 0.625 mm section collimation, 420 ms rotation time, 120 kV and 800 mA	42 months	Chi-squared test, multiple logistic regression analysis (forward conditional)	CCTA may be useful for risk stratification of coronary artery disease as added value over CACS in selected populations with low CACS who have predictors of significant NCP.

Abbreviations: ASX = asymptomatic, AUC = area under the curve analysis, CACS = coronary artery calcium score; CT = computed tomography; NRI = net reclassification index; CE = cardiac events; CPhM = Cox proportional hazard model analysis, EBCT = electron beam computed tomography; ETT = excise tolerance test; DLP = dyslipidemia, DM = diabetes mellitus; FH = family history; HT = hypertension; HCL = hypercholestrerolemia; KMA = Kaplan–Meier analysis; KMSA = Kaplan–Meier survival analysis; NR = not reported; P = prospective, R = retrospective, ROC = receiver operating characteristic curve; SPECT = single-photon emission computed tomography; SS = smoking status, SX = symptomatic; N. = number; Pts. = patients.

## Data Availability

The datasets analyzed during the current study are available from the corresponding author on reasonable request.

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
