# Peer review of "Prognostic Value of Coronary Calcium Score in Asymptomatic Individuals: A Systematic Review"

_jcm, 2022, doi:10.3390/jcm11195842_

Round 1

Reviewer 1 Report

Tramontano et al. in the manuscript "Prognostic value of Coronary Calcium Score in asymptomatic individuals: a systematic review "have attempted to prepare a systematic review summarizing reports on the prognostic role of CAC in asymptomatic patients.

A total of 45 articles were selected. Many of these (25 studies) evaluated the prognostic value of CAC score in asymptomatic subjects.

The introduction is written in an exhaustive way, appropriately introducing to the rest of the article.

The Systematic Review methodology is correct, in line with the Preferred Reporting Items for Systematic Reviews and Meta-Analyzes (PRISMA).

The obtained information is presented in a clear table and then properly discussed in the Discussion section.

Authors findings showed that the CAC score provides valuable prognostic information for predicting CAD risk in asymptomatic individuals Even though it is not a perfect study, it should be a valuable predictive tool.

As the main limitations of their study, the authors cite quite a large percentage of retrospective studies, which may in some way affect the results obtained.

Kind Regards,

Author Response

Thank you for the comment.

Reviewer 2 Report

The table that reports the characteristics of the studies is very long and covers at least 1/3 of the 25-page publication, which could become unattractive for the reader, a comparison of the most frequent among the analyzed trials could be made and obtain a table with a common point for these, since the conclusions will come out of this analysis, and I could pass this extensive table to the annexes, in general I see the document very well and it would be my only recommendation in order to make reading easier

Author Response

Thanks for your recommendation. As you suggested, we have modify the table: we have grouped the common parameters and analysis for an easier reading.

Reviewer 3 Report

In this review of the literature, the authors explore the role of the Calcium Score (CAC) as a tool for predicting the prevalence of CAD in asymptomatic patients.

The main findings of the studies included in this analysis is that CAC is an excellent tool. However, some  believe that CAC is adequate in predicting CAD only  if is associated  with other imaging methods or biomarkers. Most of the studies indicated a strong correlation between the increase in the CAC score and the occurrence of cardiac events.

The authors of the review show the usefulness of a minimally invasive diagnostic method in screening the risk of cardiac events. However, they show that the wide application of screening, especially in young people, is questionable due to radiation.

Author Response

Thank you for the comment.